# Simulation and Validation of Discrete Element Parameter Calibration for Fine-Grained Iron Tailings

**Jinxia Zhang** [1,2], **Zhenjia Chang** [1], **Fusheng Niu** [1,2,*], **Yuying Chen** [1], **Jiahui Wu** [1] **and Hongmei Zhang** [1]

1   College of Mining Engineering, North China University of Science and Technology, Tangshan 063009, China
2   Hebei Province Mining Industry Develops with Safe Technology Priority Laboratory, Tangshan 063009, China
*   Correspondence: niufusheng@ncst.edu.cn; Tel.: +86-1823-2585-555

**Abstract:** In order to improve the calculation efficiency of a discrete element EDEM (Discrete Element Method) numerical simulation software for micron particles, the particle model is linearly enlarged. At the same time, the parameters of the amplified particles were calibrated according to the Hertz-Mindlin with JKR (Johnson-Kendall-Roberts) contact model to make the amplified particles have the same particle flow characteristics as the actual particles. Actual tests were utilized to gather the angle of repose of the microfine iron tailings, which was then used as a reference value for response surface studies based on the JKR contact model from six factors connected to the fine iron tailings particles. The Plackett-Burman test was used to identify three parameters that had a significant effect on the rest angle: static friction factor; rolling friction factor; and JKR surface energy. The Box-Behnken experiment was used to establish a second-order regression model of the rest angle, and the significant parameters and the optimized parameters were: surface energy JKR coefficient 0.459; particle-particle static friction coefficient 0.393; and particle-particle dynamic friction coefficient 0.393, with a dynamic friction coefficient between particles of 0.106. By entering the parameters into the discrete element program, the angle of repose generated from the simulations was compared with the real test values, and the error was 1.56%. The contact parameters obtained can be used in the discrete element simulation of the amplified particles of fine-grained iron tailings, providing an EDEM model reference for the numerical simulation of fine-grained iron tailings particles. There is no discernible difference between the actual and simulated angles.

**Keywords:** iron tailings with fine grains; discrete element approach; angle of repose; model for second-order regression

## 1. Introduction

With the continual expansion and improvement of human civilization, the tailings created by the enormous extraction of natural resources have caused significant pollution and great harm to the environment, as well as various difficulties in terms of land and resources [1,2]. Tailings storage and tailings filling are the primary methods of managing tailings. The slurry in the mineral separation stage frequently contains a substantial quantity of water, and the high water content of tailings can readily lead to a dam failure in tailings storage. As a result, studying the flow characteristics of tailings particles in thickening equipment is an essential foundation for assuring tailings storage dependability. It has considerable guiding value for enhancing tailings usage, as well as reducing tailings storage mishaps [3,4].

Iron tailings are waste after beneficiation, and are the main component of industrial solid waste. In addition to containing a small amount of metal components, the chemical composition of iron tailings mainly contains $SiO_2$, $Al_2O_3$, $Fe_2O_3$, $CaO$, $MgO$, etc., as well as a small amount of $K_2O$, $Na_2O$ and S, P, etc. [5]. The mineral composition of iron tailings varies greatly due to the different origins and processing processes, except for the main

mineral composition of quartz, hematite, dolomite and feldspar, and other minerals such as hornblende and chlorite have unequal contents [6].

In this paper, we focus on the study of fine-grained iron tailings. The particle size of fine-grained iron tailings is small, and the number of material particles in the mineral processing and concentration equipment far exceeds the current level of computer arithmetic. The particle amplification method is a common discrete element simulation technique, and the amplified particles are re-calibrated for the coefficients of the contact model, which has important research significance in improving the computational efficiency while ensuring the accuracy of the numerical simulation [7,8]. The modeling of very viscous systems is possible thanks to the Hertz-Mindlin with JKR (Johnon-Kendall-Roberts) contact model, a cohesive contact model that contains the concept of inter-particle surface energy [9,10]. In order to demonstrate that the contact parameters obtained from calibration were used for the discrete element simulation of wheat flour amplified particles to provide a reference for them, Li Yongxiang et al. [11] used the particle contact scaling principle and gauge analysis for particle scaling in the discrete element software-based EDEM kind of JKR contact model. In order to more successfully use EDEM simulations on an industrial scale, Thomas Rosler et al. [12] investigated the scaling of the rest angle test and its impact on the calibration of DEM parameters using amplified particles; the accuracy of the calibrated values of the particle-to-particle contact parameters determines the accuracy of the computational model, according to MICHELE et al. [13], who noted that the biggest challenge in solving particle problems with EDEM (discrete element) software lies in the calibration of the model microscopic parameters. According to Ma Guangguo et al. [14], discrete element simulations were performed using the Hertz-Mindlin with JKR model to obtain three significant parameters affecting the resting angle, and response surface experiments were carried out to serve as a benchmark for the calibration of the contact parameters of shotcrete wet bodies.

In this study, Hertz-Mindlin with JKR was used as the contact model for the parameter calibration trials on fine-grained iron tailings using the discrete element analysis program EDEM (2018) [15]. The response value for scaling up the particle size of the fine-grained iron tailings and performing the parametric calibration tests was the observed resting angle. Plackett-Burman and Box-Behnken were used to create a second-order regression model of the contact parameters and rest angle of the fine-grained iron tailings. In order to show the viability of response surface experiments for calibrating discrete element particle coefficients, and to provide a reference for the EDEM (discrete element) model for the numerical simulation of fine-grained iron tailings particles, the simulated rest angle is compared with the actual rest angle.

## 2. Experiments to Calibrate Parameters

### 2.1. JKR Contact Discrete Element Model

When considerable inter-particle bonding and agglomeration takes place as a result of electrostatic forces, moisture content, etc., in materials containing moisture, such as crops, ore particles and clay, the Hertz Mindlin contact model with the JKR contact model is relevant [16]. In this research, the surface adhesion of bigger stone particles and agglomeration of fine-grained iron tailings may both be addressed using the JKR cohesion model.

The fine-grained iron tailings particles are represented by spheres in a simplified model, as shown in Figure 1a, and the surface adhesion between the particles equals the equivalent surface energy. In the JKR contact model, when particles are impacted by surface energy, the contact radius between particles changes, as illustrated in Figure 1b, and the contact radius expands. The adhesive force between various tiny particles is represented as

$$W = r_1 + r_2 - r_{12} \tag{1}$$

where $F_{\text{JKR}}$ stands for the normal elastic contact force, N; $E^*$ for the equivalent Young's modulus, Pa; $R^*$ for the equivalent radius, m; $a_2$ for the contact radius, m; $W$ for the equivalent surface energy of the contact particle, J/m$^2$; and $u$ for the normal overlap, m.

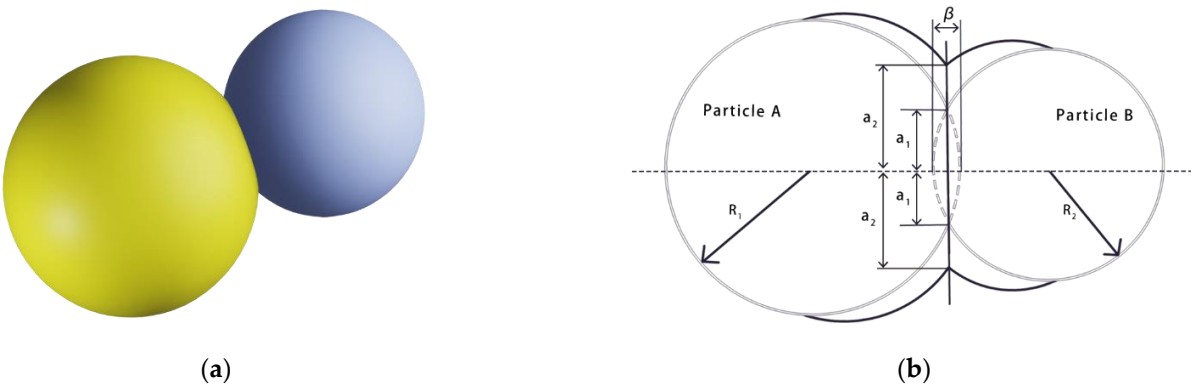

**Figure 1.** Particle model (**a**) and JKR contact theory bonding diagram between particles (**b**).

The surface energies of particle 1 and particle 2 are represented in Equation (1) by $r_1$ and $r_2$, respectively. The border energy between particle 1 and particle 2 is represented by the number $r_{12}$.

The boundary energy between particles is zero, and the surface energy values of various particles are the same: currently, $r_{12} = 0$ and $r_1 = r_2 = r$. As a result, $W = 2r$ is the equation for the cohesion between similar particle types [17]. In (2) and (3), respectively, the normal elastic contact force JKR and the normal overlap u of the particles are displayed:

$$F_{JKR} = -2(2\Pi W E^* a_2^3)^{\frac{1}{2}} + \frac{4E^* a_2^3}{3R^*} \tag{2}$$

$$u = \frac{a_2^2}{R^*} - \sqrt{\frac{2\Pi W a_2}{E^*}} \tag{3}$$

When $W = 2r$ is introduced into Equations (2) and (3) and the particle types are the same, the normal elastic contact force $F_{JKR}$ normal to the overlap $u$ can be written as follows

$$F_{JKR} = -4\sqrt{\Pi r E^* a_2^3} + \frac{4E^* a_2^3}{3R^*} \tag{4}$$

$$u = \frac{a_2^2}{R^*} - 2\sqrt{\frac{\Pi W a_2}{E^*}} \tag{5}$$

where $a_2$ is the radius of the contact surface following the collision of the two particles, $r$ is the surface energy of the contacting particles, $E^*$ is the modulus of elasticity and $R^*$ is the equivalent contact radius.

To increase simulation accuracy when scaling the particle size, the adjusted particle size's coefficients must be calibrated so that the scaled particles have the same attributes as the original particles [18]. To determine the scaling factors between specific physical quantities between the physical model of the original system and the scaled model, Feng et al. [19] employed a straightforward method

$$\overline{Q} = \varepsilon_q \times Q \tag{6}$$

where $Q$ is any parameter in the physical system; $\varepsilon_q$ is the scale factor; and $\overline{Q}$ is any parameter in the scaling system.

According to the pertinent literature [20], as the square of the scaled particle size increases, so too does the contact force between particles, and as the particle radius increases, so does the interparticle contact surface area. Since no reference range was stated for the Hertz-Mindlin with JKR contact model species, it was established using coefficient calibration studies.

### 2.2. Sizing for Discrete Element Models

This research suggests an expansion strategy based on the particle size of fine-grained iron tailings in an effort to maximize computing efficiency [21]. The larger particles are meant to serve as a reference EDEM (discrete element) model for the numerical simulation of fine-grained iron tailings based on the experimental concentrator as a model. The enlarged particles of various sizes are shown in Figure 2b to demonstrate that the overall area of the particles is the same as the original particle area in terms of percentage, ensuring that the enlarged particle calculation results have high accuracy. Figure 2a depicts the distribution of various particle sizes within a given area, with different colors representing particles of different sizes.

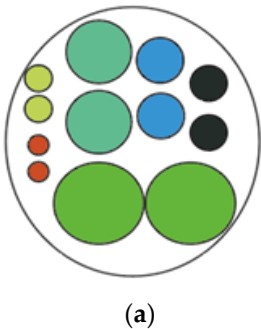   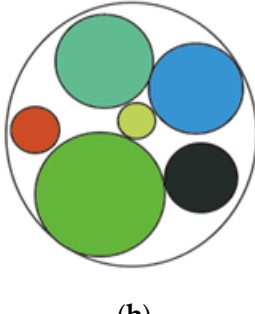

(**a**)                  (**b**)

**Figure 2.** Theoretical diagram of original particle size distribution (**a**) and amplified particle size distribution (**b**).

Where: $R_a$ is the particle size, mm; $T$ is the device volume, L; $\overline{\rho}$ is the original particle average size, mm; $X_n$ is the enlarged large particle size, mm; and $a_n$ is the particle radius with different occupancy ratio, mm.

The average particle size $R_a$ of the amplified particles is first calculated while performing particle size analysis

$$R_a = \frac{T}{G} \tag{7}$$

where $G$ is the maximum point at which the equipment can hold the particles.

To determine the particle amplification factor k for various particle sizes t, $R_a$ is introduced into Equation (8):

$$k = \frac{R_a}{\overline{\rho}} \tag{8}$$

The amplified discrete element particles $X_n$ are finally made:

$$\frac{4}{3}\pi X_n^3 = k\frac{4}{3}\pi a_n^3 \tag{9}$$

For trials to calibrate coefficients, the resulting particle distribution was entered into the discrete element program.

### 2.3. Determination of the Angle of Repose

The rest angle tester shown in Figure 3 was designed according to the widely used rest angle determination method, which mainly consists of an iron frame table, a 1 mm mm funnel, and a rectangular chassis. The resulting resting angle was image processed to measure the most accurate resting angle possible and to eliminate interfering factors. First, using the Python language, the resulting (Figure 4a) was grayscale processed to obtain (Figure 4b); set the threshold to 100, 150, 200 to select the optimal observation image for binarization, the processing process (Figure 4c) [22]; using Illustrator software to contour curve processing of the image after the resulting curve in OriginPro (2018) in the image digitizing tool for coordinate identification (Figure 4d) [23], the identification data fit well with the curve and are representative; at the same time, the identification data were linearly

fitted (Figure 4e), *Adjusted R squared* = 0.99852, the model fit is good and representative; finally, the rest angle was judged according to the slope of the curve, and the rest angle was 45.81°. The response value was determined for the later calibration experiment.

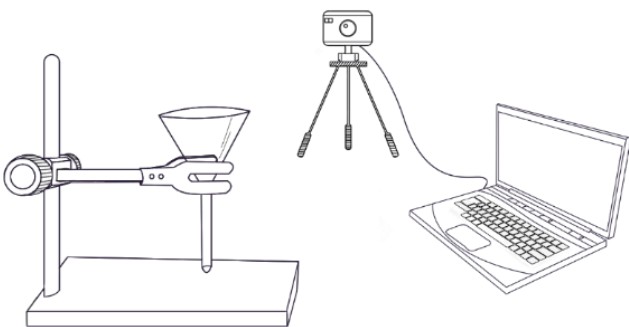

**Figure 3.** Angle of repose meter.

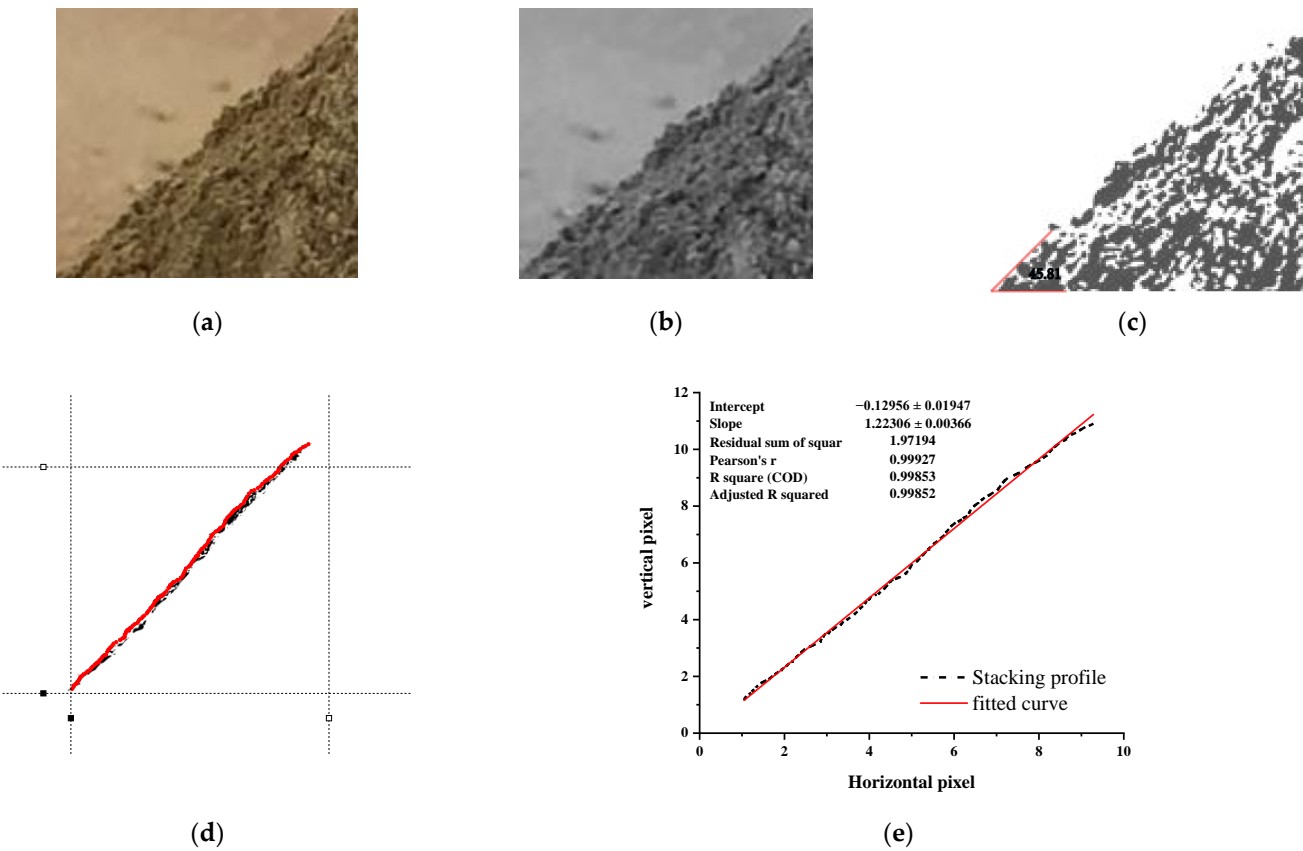

**Figure 4.** Response value image processing.

### 2.4. Particle Modeling with Discrete Elements

In order to replicate the real working circumstances as closely as feasible, discrete element simulation was performed in EDEM. The model was reduced to include spherical particles to increase simulation efficiency by examining the actual iron tailings particles (Figure 5a), which may be used to acquire particle microscope pictures of the individual particles [24] (Figure 5b).

Figure 6 displays the actual particle size distribution of iron tailings. The average particle size of iron tailings is 24.15 μm, according to the real particle size distribution of the tailings.

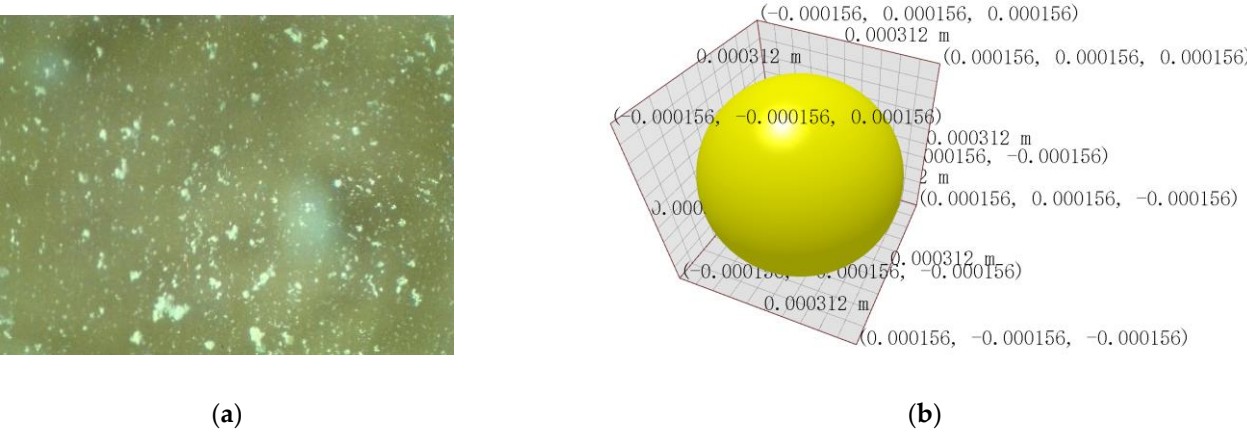

(**a**)                                (**b**)

**Figure 5.** Particle microscope image (**a**) and discrete element particle image (**b**).

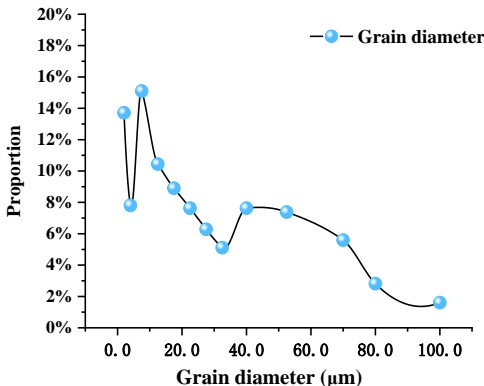

**Figure 6.** Tailings true particle size distribution map.

The numerical simulation of the expanded particles was carried out using the concentrator as the goal vessel and a limiting value of 500,000 particles [25]. In Figure 7, the average particle size is 1.959708 mm.

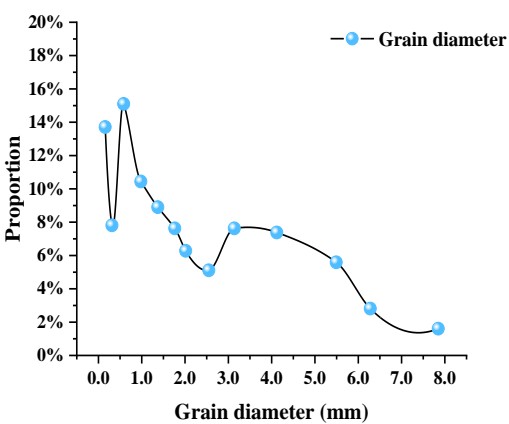

**Figure 7.** Enlarged size distribution of particles.

The particle factory was set up in EDEM, the particle production technique was set to Dynamic factory and the generation rate was set to 5 kg/s. The pile-up model was created using ANSYS proprietary modelling program Space Claim. By manually shutting down the particle factory once a predetermined number of particles had been produced and setting the time step to 30%, the numerical simulation's accuracy and efficiency were guaranteed. The computation period of 10S was designed to be significantly longer than the actual simulation duration in order to create a stable particle state, and a time step

of 30% was set to assure the effectiveness and accuracy of the numerical simulation. To clearly illustrate the process of particle accumulation inside the container, as shown in the simulation flowchart in Figure 8, the different colours of the particles represent the speed during the motion, with red representing the fastest and blue-green the next fastest. After a predetermined number of particles are generated from the top, as in Figure 8a, the particles fall freely, as in Figure 8b, due to the acceleration caused by gravity, the particles at the bottom fall at the fastest speed as in Figure 8c and finally come into contact with the stacking surface after forming an angle of repose.

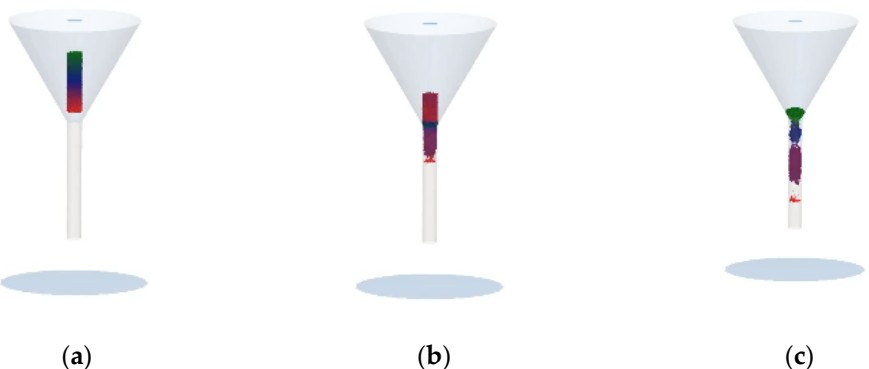

(a)                                   (b)                                   (c)

**Figure 8.** Schematic diagram of discrete element repose simulation flow.

## 3. Designing and Analyzing Studies for Parameter Calibration

Plackett-Burman and Box-Behnken experiments are combined in the parameter calibration experiment. Plackett-Burman analysis [26] reveals that the factor that needs to be calibrated the most has the greatest impact on the angle of repose; the relationship model between the angle of repose and the significant parameters is then obtained by analyzing the significant factors and carrying out the Box-Behnken experiment, and the best parameters are then examined. Finally, the accuracy is examined by comparison with the actual particle angle of repose and by confirming the viability [27].

### 3.1. Plackett-Burman Experimental Design Importance

Table 1 shows the initial determination of the coefficient table and the range of values based on the particle coefficients of iron tailings, with reference to the literature [28] and the "GMEE (Generic EDEM material model database)" database that comes with the discrete element software EDEM, and with reference to the properties of fine-grained iron tailings. Table 2 displays the Plackett-Burman test findings.

**Table 1.** Parameter calibration test of discrete element method calibration parameters.

| Simulation Parameters | | Level | |
|---|---|---|---|
| | | Low Level | High Level |
| Particle Poisson's ratio | A | 0.3 | 0.5 |
| Coefficient of shear elasticity (pa) | B | $2.40 \times 10^9$ | $2.40 \times 10^{10}$ |
| JKR surface energy coefficient (J/m$^2$) | C | 0.3 | 0.6 |
| Collision recovery factor (particles) | D | 0.1 | 0.3 |
| Coefficient of static friction (particles) | E | 0.3 | 0.5 |
| Coefficient of dynamic friction (particles) | F | 0.08 | 0.12 |

The results of the significance analysis of the parameters obtained from the Plackett-Burman test [29] are shown in Table 3. The final regression equation for the real factor produced the following result, using the resting angle R as the response value: $R = 32.59 - 0.2275A - 0.0133B + 3.32C + 0.4650D + 3.10E + 2.73F + 0.5212AC + 0.7162AE$, $R^2 = 0.9922$, demonstrating that the regression equation model suited the data well, and was a representative regression equation [30]. Where $R_{adj} = 0.9829$, this indicated that the model applies

to 98.29% of the effect values. The ANOVA results showed that the results for JKR surface energy coefficient (C), particle-particle static friction coefficient (E) and particle-particle dynamic friction coefficient (F) had a significant effect ($p < 0.05$) on the particle resting angle (R), while the other factors were not significant.

**Table 2.** Design and results of Plackett-Burman test.

| Serial Number | A | B (pa) | C (J/m³) | D | E | F | Repose Angle (°) |
|---|---|---|---|---|---|---|---|
| 1 | 0.50 | $2.40 \times 10^{10}$ | 0.3 | 0.3 | 0.50 | 0.12 | 35.41 |
| 2 | 0.30 | $2.40 \times 10^{10}$ | 0.6 | 0.1 | 0.50 | 0.12 | 40.29 |
| 3 | 0.50 | $2.40 \times 10^{9}$ | 0.6 | 0.3 | 0.30 | 0.12 | 35.53 |
| 4 | 0.30 | $2.40 \times 10^{10}$ | 0.3 | 0.3 | 0.50 | 0.08 | 30.12 |
| 5 | 0.30 | $2.40 \times 10^{9}$ | 0.6 | 0.1 | 0.50 | 0.12 | 40.25 |
| 6 | 0.30 | $2.40 \times 10^{9}$ | 0.3 | 0.3 | 0.30 | 0.12 | 30.94 |
| 7 | 0.50 | $2.40 \times 10^{9}$ | 0.3 | 0.1 | 0.50 | 0.08 | 29.18 |
| 8 | 0.50 | $2.40 \times 10^{10}$ | 0.3 | 0.1 | 0.30 | 0.12 | 27.03 |
| 9 | 0.50 | $2.40 \times 10^{10}$ | 0.6 | 0.1 | 0.30 | 0.08 | 29.18 |
| 10 | 0.30 | $2.40 \times 10^{10}$ | 0.6 | 0.3 | 0.30 | 0.08 | 30.96 |
| 11 | 0.50 | $2.40 \times 10^{9}$ | 0.6 | 0.3 | 0.50 | 0.08 | 37.85 |
| 12 | 0.30 | $2.40 \times 10^{9}$ | 0.3 | 0.1 | 0.30 | 0.08 | 24.35 |

**Table 3.** Plackett-Burman test parameter significance analysis.

| Factors | Sum of Squares | F-Value | *p*-Value | Effect |
|---|---|---|---|---|
| Models | 293.48 | 106.54 | <0.0001 | |
| A | 0.6211 | 1.35 | 0.2973 | −0.2275 |
| B | 2.18 | 4.74 | 0.0814 | −0.425833 |
| C | 114.27 | 248.88 | <0.0001 | 3.08583 |
| D | 9.24 | 20.13 | 0.0065 | 0.8775 |
| E | 102.73 | 223.74 | <0.0001 | 2.92583 |
| F | 64.45 | 140.37 | <0.0001 | 2.3175 |
| Residual | 2.30 | | | |
| Total deviation | 295.78 | | | |

Pareto Figure 9 was produced as a consequence of additional model data analysis. Figure 9 shows the important variables, and in accordance with the data table, only these three parameters with substantial impacts were taken into account in the ensuing Box-Behnken experiments [31]. The picture also shows the impacts of each component, both favorable and unfavorable, on the response values, serving as a guide for the following calibration of the coefficient parameters.

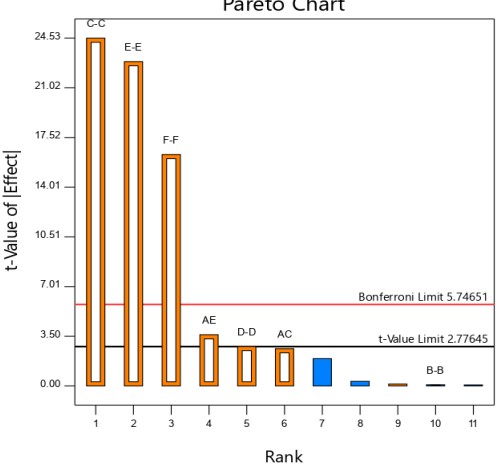

**Figure 9.** Pareto chart diagram.

### 3.2. Box-Behnken Response Surface Analysis

Based on the Plackett-Burman test findings, a Box-Behnken test design with three degrees of significance was conducted, and three center points were chosen to evaluate the error [32]. Table 4 displays the results of the Box-Behnken test.

**Table 4.** Box-Behnken experimental design and results.

| Serial Number | C (J/m$^2$) | E | F | Repose Angle (°) |
|---|---|---|---|---|
| 1 | 0.30 | 0.40 | 0.08 | 34.36 |
| 2 | 0.45 | 0.40 | 0.10 | 45.88 |
| 3 | 0.30 | 0.40 | 0.12 | 34.11 |
| 4 | 0.45 | 0.30 | 0.08 | 33.32 |
| 5 | 0.45 | 0.40 | 0.10 | 44.13 |
| 6 | 0.60 | 0.40 | 0.08 | 29.19 |
| 7 | 0.45 | 0.40 | 0.10 | 44.12 |
| 8 | 0.45 | 0.40 | 0.10 | 45.34 |
| 9 | 0.45 | 0.30 | 0.12 | 39.45 |
| 10 | 0.30 | 0.30 | 0.10 | 28.13 |
| 11 | 0.30 | 0.50 | 0.10 | 30.13 |
| 12 | 0.60 | 0.40 | 0.12 | 35.13 |
| 13 | 0.45 | 0.40 | 0.10 | 46.34 |
| 14 | 0.60 | 0.30 | 0.10 | 29.23 |
| 15 | 0.45 | 0.50 | 0.12 | 34.45 |
| 16 | 0.60 | 0.50 | 0.10 | 25.19 |
| 17 | 0.45 | 0.50 | 0.08 | 33.34 |

The Box-Behnken test model ANOVA results are shown in Table 5. According to the results in Table 5, it can be seen that the $p < 0.0001$ for the fitted model; and the $p$-values for the JKR surface energy coefficient (C), static friction coefficient (E), dynamic friction coefficient (F), JKR surface energy coefficient × static friction coefficient (CE), JKR surface energy-rolling friction coefficient (CF), static friction coefficient x dynamic friction coefficient (EF) and the quadratic terms for each parameter are <0.05, indicating that the individual parameters with a resting angle were significant, as well as the validity of the regression model. The misfit term $p = 0.4458 > 0.05$ indicates that the model is good and no bending misfit occurs. Coefficient of determination $R^2 = 0.9884$; and the corrected coefficient of determination $R^2_{adj} = 0.9736$. The predictive coefficient of determination $R^2_{pre} = 0.9064$, indicating that the model is a true representation of the actual situation [33]. The test precision *Adep Precision* = 24.6778, indicating that the model has good accuracy.

**Table 5.** Box-Behnken experimental model ANOVA.

| Source of Variance | Sum of Squares | Freedom | Mean Square | F-Value | *p*-Value |
|---|---|---|---|---|---|
| Models | 637.64 | 9 | 70.85 | 66.47 | <0.0001 |
| C | 272.84 | 1 | 272.84 | 255.99 | <0.0001 |
| E | 13.47 | 1 | 13.47 | 12.64 | 0.0093 |
| F | 18.30 | 1 | 18.30 | 17.17 | 0.0043 |
| C × E | 22.09 | 1 | 22.09 | 20.73 | 0.0026 |
| C × F | 7.18 | 1 | 7.18 | 6.74 | 0.0356 |
| E × F | 6.30 | 1 | 6.30 | 5.91 | 0.0453 |
| C$^2$ | 29.09 | 1 | 29.09 | 27.29 | 0.0012 |
| E$^2$ | 205.64 | 1 | 205.64 | 192.94 | <0.0001 |
| F$^2$ | 38.75 | 1 | 38.75 | 36.35 | 0.0005 |
| Residual | 7.46 | 7 | 1.07 | | |
| Lack of fit | 3.38 | 3 | 1.13 | 1.10 | 0.4458 |
| Pure error | 4.09 | 4 | 1.02 | | |
| Sum | 645.10 | 16 | | | |
| $R^2 = 0.9884$ | $R^2_{adj} = 0.9736$ | $R^2_{pre} = 0.9064$ | *Adep Precision* = 24.6778 | | |

The regression equation obtained from the Box-Behnken test:

$$R = -219.24050 + 162.07333C + 679.35500E + 1642.37500F - 156.66667CE + 446.66667CF - 627.50000EF - 116.82222C^2 - 698.85000E^2 - 7583.75000F^2$$

### 3.3. Regression Model Interaction Effect Analysis

Conversely, the further away from a circle the contour plot deviates from a circular interaction response, and vice versa [34]. The interaction between the two independent variables impacts the response value to a greater or lesser extent depending on the slope of the response surface plot; conversely, the flatter the interaction, the less significant the effect.

As seen in Figure 10a, the response surface plot and contour plot of the JKR surface energy coefficient and static friction coefficient are generated in Figure 10 when the fixed rolling friction coefficient F is 0.10. The particle resting angle exhibits an increasing trend as the JKR surface energy coefficient C rises. The particles' resting angle height rises from 28.13° to 45.66° when the static friction coefficient is 0.4, and the JKR surface energy coefficient increases from 0.30 J/m² to 0.60 J/m². According to the JKR contact model, an increase in the surface energy coefficient causes an increase in the cohesive forces that exist between particles. These forces can cause particles to adhere to one another and form new particle clusters, which causes the angle of repose to increase, as the particles at the top of the angle of repose are adsorbed and difficult to slide off when stacking is done. A strong interaction between the JKR surface energy coefficient C and the static friction coefficient E between the particles can be taken into account in the contour plot of Figure 10b.

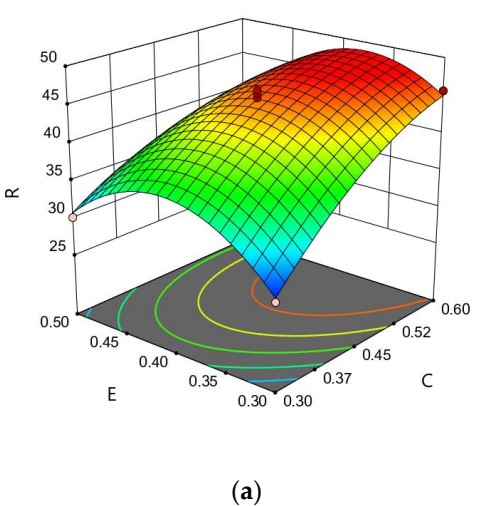

(**a**)

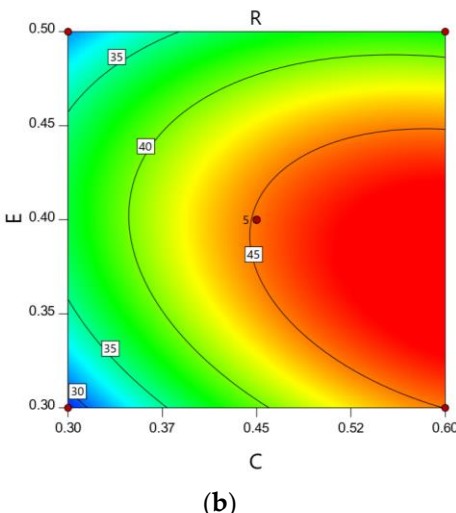

(**b**)

**Figure 10.** Response surface diagram (**a**) and contour diagram (**b**) of the influence of JKR surface energy coefficient and static friction coefficient on repose angle.

The surface plot and contour plot of the influence of surface energy and rolling friction coefficient on the resting angle of JKR are obtained, as shown in Figure 11a, where the fixed static friction coefficient E is equal to 0.40. With an increase in the rolling friction coefficient, the particles' angle of repose increases. This is because the gap between particles shrinks as the rolling coefficient between them rises, and the larger the rolling coefficient, the smaller the repulsion between particles and the higher the particle stacking angle will be, at which point the particle rolling friction coefficient positively influences the resting angle [35]. This is because when the rolling friction coefficient reaches a particular level, it will result in a decrease in the cohesiveness between the particles, which will impact the resting angle of the particles. The JKR surface energy C and rolling friction coefficient F contours are not circular in the contour plot in Figure 11b, and the JKR surface energy coefficient and rolling friction coefficient interact with one another between the particles.

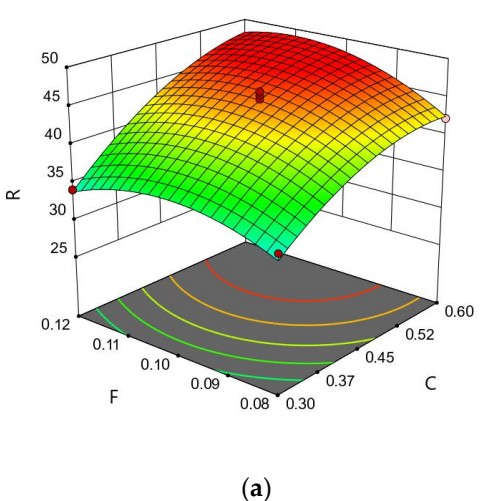
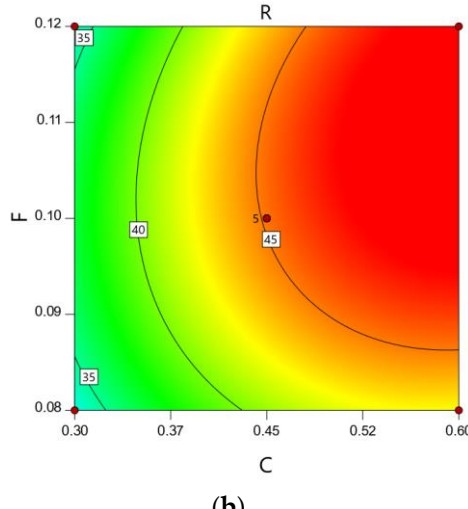

(**a**)  (**b**)

**Figure 11.** Influence of JKR surface energy and rolling friction coefficient on angle of repose surface diagram (**a**) and contour diagram (**b**).

The response surface plots of the static friction coefficient E and kinetic friction coefficient F obtained in Figure 12a all have blatantly steep surfaces, indicating that the interaction effect on the response value rest angle is significant in all degrees when the JKR surface energy coefficient = 0.45. The interaction is significant, which is consistent with the variance results, as shown by the contours of the static friction coefficient E and dynamic friction coefficient F in Figure 12b, which all deviate from the circle.

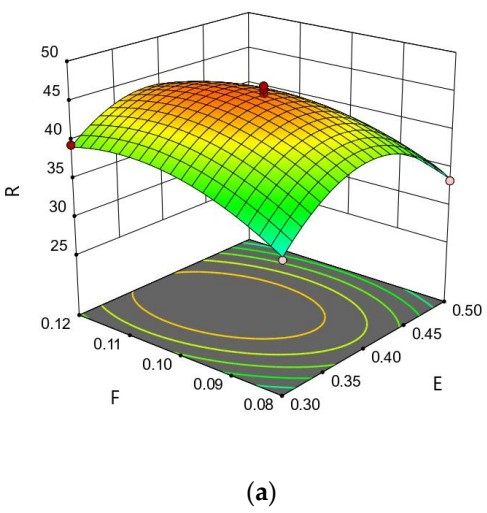
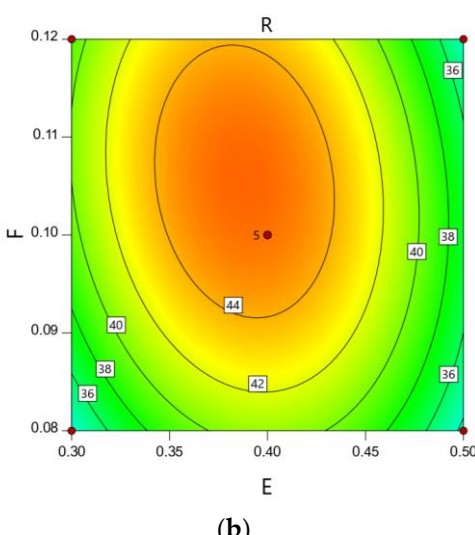

(**a**)  (**b**)

**Figure 12.** Surface diagram (**a**) and contour diagram (**b**) of the influence of JKR surface energy and rolling friction coefficient on repose angle.

## 4. Determination of the Optimal Combination of Parameters and Validation of the Simulation

Targeting the real resting angle of fine-grained iron tailings using Design-Expert 11 software, the JKR surface energy is 0.459, the static friction coefficient is 0.393 and the dynamic friction coefficient is 0.106. By minimizing the error in the angle of repose between the simulation and the experiment, the optimized regression equation may be solved. The resting angle was 44.81°, which is 2.18% different from the projected value, and the error is acceptable. The findings of the response surface parameters were entered into the discrete element software for comparison tests.

The optimal set of parameters was used to execute the rest angle simulation test, and Figure 13 compares the results of the simulation and physical test. In order to verify the accuracy of the data obtained from the response surface and the actual data, the virtual simulation of the particle stacking process was performed ten times in EDEM (2018) to observe the particle angle and record the experiment, while the basic operation of the rest angle measuring instrument Figure 3 was performed for the stacking experiment under the same conditions, as shown in Figure 14. The performance of the point line diagram is basically consistent, and the error exists within the acceptable range. The mean value of the rest angle obtained from the simulation test was 45.823°, with an error of 1.56% from the actual value of 45.119°, indicating that there was no significant difference between the simulation results and the real test value.

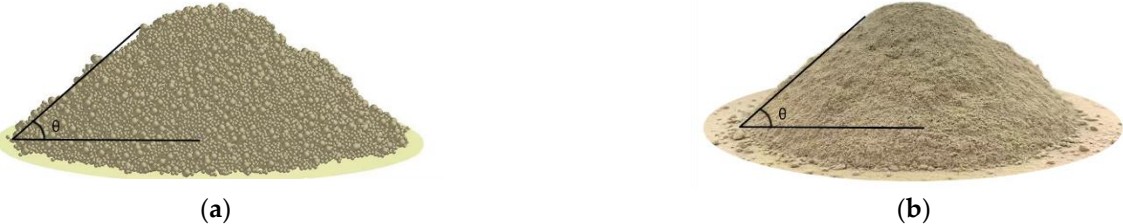

(**a**)  (**b**)

**Figure 13.** Simulation stacking (**a**) and actual stacking (**b**).

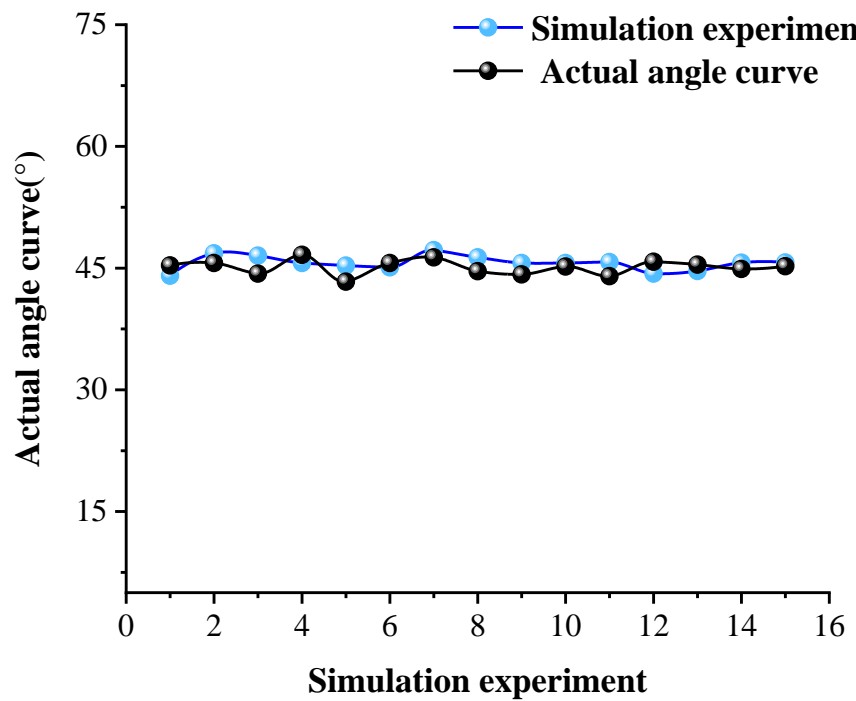

**Figure 14.** Simulation experiment and actual angle dot plot.

## 5. Conclusions

(1) The computational performance of the numerical simulation was improved by increasing the discrete element of fine-grained iron tailings' particle size by 1.959708 mm, with an average particle size of 24.15 um, and using 500,000 particles as the maximum.

(2) The contact characteristics of the amplified particles were calibrated using the JKR contact model in discrete elements. The Plackett-Burman tests were used to determine the factors that significantly affect the resting angle of the amplified particles of microfine-grained iron tailings. These factors included the surface energy JKR coefficient, particle-particle static friction coefficient and particle-particle dynamic friction coefficient.

(3)　The Box-Behnken test revealed that, in contrast to the simulated particle rest angle of 44.81° for fine-grained iron tailings particles at 0.459, 0.393 and 0.106, respectively, the relative error of the surface energy JKR coefficient, particle-particle static friction coefficient and particle-particle kinetic friction coefficient in the EDEM discrete element software was only 2.18%; this proves the viability of response surface experiments for the discrete element particle system. It was shown that it was possible to calibrate the particle coefficients for discrete elements.

(4)　The best experimentally obtained parameters were entered into discrete element software, where the mean resting angle was calculated to be 45.823°. This was compared to the mean angle from physical experiments, which was 45.119°, and the error was calculated to be 1.56%, which was not significantly different. This proves that the contact parameters obtained from the particle size scaling coefficient calibration trials satisfy the numerical simulation's requirements, and serve as a reference for the discrete element model used to simulate the numerical behavior of fine-grained iron tailings particles.

**Author Contributions:** J.Z. and Z.C. designed the experiments, Z.C. performed the experiments, Y.C. and J.W. analyzed the data, Z.C. and H.Z. wrote the paper and F.N. improved the paper. All authors have read and agreed to the published version of the manuscript.

**Funding:** This research was funded by the National Natural Science Foundation of China (Grant No. 51874135; No. 51904106), the Natural Science Foundation of Hebei Province (Grant No. E2021209015; No. E2022209108), and key projects of Hebei Provincial Department of Education (Grant No. ZD2022059) and Hebei Provincial High-Level Talents Funding Project (Grant No. B20221005).

**Institutional Review Board Statement:** Not applicable.

**Informed Consent Statement:** Not applicable.

**Data Availability Statement:** Data available on request due to restrictions eg privacy or ethical.The data presented in this study are available on request from the corresponding author. The data are not publicly available due to As the data needs.

**Conflicts of Interest:** The authors declare no conflict of interest.

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
