# Peer review of "Simulation and Validation of Discrete Element Parameter Calibration for Fine-Grained Iron Tailings"

_minerals, doi:10.3390/min13010058_

Round 1

Reviewer 1 Report

Thank you for your interesting and valuable manuscript. The manuscript is well-written, and the methodology and results are innovative.  However, some details need to be revised.

-Some figures are blurry and unclear.

-The font of variable symbols should be in italic.

-The regression equations are important results of the manuscript and should be listed separately rather than in the paragraphs of the text.

Author Response

请参阅附件

Reviewer 2 Report

In this manuscript, the author investigated the calculation and validation of discrete element parameter calibration for fine particle, and obtained contact parameters based on the JKR contact model, which provided an EDEM (discrete element) model reference for the numerical simulation of fine iron-bearing tailings. This is an interesting work. I suggested that it can be accepted after major revision. Notably, the following issues must be addressed.

1. For Abstract, the first appearance of abbreviation needs to indicate the full name, e.g., EDEM and JKR.

2. In Introduction, a description of the mineral composition of iron tailings should be added, please add the references (Minerals Engineering, 2021, 173, 107191; Resources, Conservation and Recycling, 2021, 172, 105680).

3. Line 60, "MA According to Guangguo et al. [12]", what is "MA", should it be deleted?

4. Line 96, what is FJKR?

5. In this manuscript, for all the variables in the formula, their units must be specified.

6. Figure 1 is not clear, and thus the author should revise it.

7. Line 122, for "Ra", R should be italicized and a should be subscripted.

8. Tables 2, 4, the unit of Repose angle is "°", rather than "R°", please correct.

9. for Fig. 6, "um" should be corrected as "μm".

10. for Fig. 14, the unit of Angle should be added.

Reviewer 3 Report

This excellent manuscript investigates simulation and validation of discrete element parameter calibration for fine-grained iron tailings. The obtained conclusions are obvious innovation and interested for readers. The authors give sufficient literature review and attention to detail. And this paper contains in-depth and comprehensive scientific analysis. The title and abstract reflect the content of this paper. Therefore, I strongly recommend that this manuscript can be accepted for publication in Minerals after the following minor revision:

1The word major is often overused. Consider usinga more specific synonym to improve the sharpnessof your writing

2The spelling of Modelling is a non-American variantFor consistency, consider replacing it with the American English spelling

3Preposition errors occur in Section 1.3

4The curve format in Figure 6 is different from that in Figure 7

5Chart data should be uniformly retained exact bits

6The author proposes to take the total number of particles of 500,000 as the limit value. How to judge this value?

7There are some errors in the composition of the article, which should be properly typeset in the later stage

8Do more control experiments to make the simulation results more convincing

9The English needs mandatory editing.

Round 2

Reviewer 2 Report

This manuscript has been improved significantly and I think it can be accepted for publication in its current form.